# Learning Binary Trees via Sparse Relaxation

## Abstract

One of the most classical problems in machine learning is how to learn binary trees that split data into useful partitions. From classification/regression via decision trees to hierarchical clustering, binary trees are useful because they (a) are often easy to visualize; (b) make computationally-efficient predictions; and (c) allow for flexible partitioning. Because of this there has been extensive research on how to learn such trees that generally fall into one of three categories: 1. greedy node-by-node optimization; 2. probabilistic relaxations for differentiability; 3. mixed-integer programs (MIP). Each of these have downsides: greedy can myopically choose poor splits, probabilistic relaxations do not have principled ways to prune trees, MIP methods can be slow on large problems and may not generalize. In this work we derive a novel sparse relaxation for binary tree learning. By deriving a new MIP and sparsely relaxing it, our approach is able to learn tree splits and tree pruning using argmin differentiation. We demonstrate how our approach is easily visualizable and is competitive with current tree-based approaches in classification/regression and hierarchical clustering.

## 1 Introduction

Learning discrete structures from unstructured data is extremely useful for a wide variety of real-world problems (Gilmer et al., 2017; Kool et al., 2018; Yang et al., 2018). One of the most computationally-efficient, easily-visualizable discrete structures that are able to represent complex functions are *binary trees*. For this reason, there has been a massive research effort on how to learn such binary trees since the early days of machine learning (Payne & Meisel, 1977; Breiman et al., 1984; Bennett, 1992; Bennett & Blue, 1996). Learning binary trees has historically been done in one of three ways. The first is via *greedy optimization*, which includes popular decision-tree methods such as classification and regression trees (CART) (Breiman et al., 1984) and ID3 trees (Quinlan, 1986), among many others. These methods optimize a splitting criterion for each tree node, based on the data routed to it. The second set of approaches are based on *probabilistic relaxations* (İrsoy et al., 2012; Yang et al., 2018). The idea is to optimize all splitting parameters at once via gradient-based methods, by relaxing hard branching decisions into branching probabilities. The third approach optimizes trees using *mathematical programming* (MIP) (Bennett, 1992; Bennett & Blue, 1996). This jointly optimizes all continuous and discrete parameters to find globally-optimal trees.[1]

Each of these approaches have clear shortcomings. First, greedy optimization is clearly suboptimal: tree splitting criteria are even intentionally crafted to be different than the global tree loss, as the global loss does not encourage tree growth (Breiman et al., 1984). Second, probabilistic relaxations: (a) are rarely sparse, so inputs probabilistically contribute to branches they would never visit if splits are mapped to hard decisions; (b) they do not have principled ways to prune trees, as the distribution over pruned trees is often intractable. Third, MIP approaches, while optimal, are only computationally tractable on training datasets with thousands of inputs (Bertsimas & Dunn, 2017), and do not have well-understood out-of-sample generalization guarantees.

In this paper we present a new approach to binary tree learning based on sparse relaxation and argmin differentiation. Our main insight is that by quadratically relaxing an MIP that learns the discrete parameters of the tree (input traversal and node pruning), we can differentiate through it to simultaneously learn the continuous parameters of splitting decisions. This allows us to leverage the superior generalization capabilities of stochastic gradient optimization to learn splits, and gives

---

[1]Here we focus on learning single trees instead of tree ensembles; our work easily extends to ensembles.

a principled approach to learning tree pruning. Further, we can derive customized algorithms to compute the forward and backward passes through this program that are much more efficient than generic approaches (Amos & Kolter, 2017). We demonstrate that (a) in classification/regression our method, which learns a single tree and a classifier on top of it, is competitive with greedy, probabilistic, MIP-based tree methods, and even powerful ensemble methods; (b) in hierarchical clustering we match or improve upon the state-of-the-art.

## 2 RELATED WORK

The paradigm of *binary tree learning* has the goal of finding a tree that iteratively splits data into meaningful, informative subgroups, guided by some criterion. Binary tree learning appears in a wide variety of problem settings across machine learning. We briefly review work in two learning settings where latent tree learning plays a key role: 1. *Classification/Regression*; and 2. *Hierarchical clustering*. Due to the generality of our setup (tree learning with arbitrary split functions, pruning, and downstream objective), our approach can be used to learn trees in any of these settings. Finally, we detail how parts of our algorithm are inspired by recent work in isotonic regression.

**Classification/Regression.** Decision trees for classification and regression have a storied history, with early popular methods that include classification and regression trees (CART; Breiman et al., 1984), ID3 (Quinlan, 1986), and C4.5 (Quinlan, 1993). While powerful, these methods are greedy: they sequentially identify 'best' splits as those which optimize a split-specific score (often different from the global objective). As such, learned trees are likely sub-optimal for the classification/regression task at hand. To address this, Carreira-Perpinán & Tavallali (2018) proposes an alternating algorithm for refining the structure and decisions of a tree so that it is smaller and with reduced error, however still sub-optimal. Another approach is to probabilistically relax the discrete splitting decisions of the tree (İrsoy et al., 2012; Yang et al., 2018; Tanno et al., 2019). This allows the (relaxed) tree to be optimized *w.r.t.* the overall objective using gradient-based techniques, with known generalization benefits (Hardt et al., 2016; Hoffer et al., 2017). Variations on this approach aim at learning tree ensembles termed 'decision forests' (Kontschieder et al., 2015; Lay et al., 2018; Popov et al., 2019). The downside of the probabilistic relaxation approach is that there is no principled way to prune these trees as inputs pass through all nodes of the tree with some probability. A recent line of work has explored mixed-integer program (MIP) formulations for learning decision trees. Motivated by the billion factor speed-up in MIP in the last 25 years, Rudin & Ertekin (2018) proposed a mathematical programming approach for learning provably optimal decision lists (one-sided decision trees; Letham et al., 2015). This resulted in a line of recent follow-up works extending the problem to binary decision trees (Hu et al., 2019; Lin et al., 2020) by adapting the efficient discrete optimization algorithm (CORELS; Angelino et al., 2017). Related to this line of research, Bertsimas & Dunn (2017) and its follow-up works (Günlük et al., 2018; Aghaei et al., 2019; Verwer & Zhang, 2019; Aghaei et al., 2020) phrased the objective of CART as an MIP that could be solved exactly. Even given this consistent speed-up all these methods are only practical on datasets with at most thousands of inputs (Bertsimas & Dunn, 2017) and with non-continuous features. Further, the out-of-sample generalizability of these approaches is not well-studied, unlike stochastic gradient descent learning.

**Hierarchical clustering.** Compared to standard flat clustering, hierarchical clustering provides a structured organization of unlabeled data in the form of a tree. To learn such a clustering the vast majority of methods are greedy and work in one of two ways: 1. *Agglomerative*: a 'bottom-up' approach that starts each input in its own cluster and iteratively merges clusters; and 2. *Divisive*: a 'top-down' approach that starts with one cluster and recusively splits clusters (Zhang et al., 1997; Widyantoro et al., 2002; Krishnamurthy et al., 2012; Dasgupta, 2016; Kobren et al., 2017; Moseley & Wang, 2017). These methods suffer from similar issues as do greedy approaches to tree learning for classification/regression: they may be sub-optimal for optimizing the overall tree. Further they are often computationally-expensive due to their sequential nature. Inspired by approaches for classification/regression, recent work has designed probabilistic relaxations for learning hierarchical clusterings via gradient-based methods (Monath et al., 2019).

Our work takes inspiration from both the MIP-based and gradient-based approaches. Specifically, we frame learning the discrete tree parameters as an MIP, which we sparsely relax to allow continuous parameters to be optimized by argmin differentiation methods.

**Argmin differentiation.** Solving an optimization problem as a differentiable module within a parent problem tackled with gradient-based optimization methods is known as argmin differentiation, a particular instance of bi-level optimization (Gould et al., 2016). This situation arises in as diverse scenarios as hyperparameter optimization (Pedregosa, 2016), meta-learning (Rajeswaran et al., 2019), or structured prediction (Stoyanov et al., 2011; Domke, 2013; Niculae et al., 2018). General algorithms for quadratic Amos & Kolter (2017) and disciplined convex programming (Section 7, Amos, 2019; Agrawal et al., 2019a;b) have been given, as well as expressions for more specific cases like isotonic regression (Djolonga & Krause, 2017). By taking advantage of the very specific structure of the decision tree induction problem, we obtain a direct, efficient algorithm.

**Isotonic regression.** Also called monotonic regression, isotonic regression (Barlow et al., 1972) constrains the regression function to be non-decreasing/non-increasing. This is useful if one has prior knowledge of such monotonicity (*e.g.*, the mean temperature of the climate is non-decreasing). A classic algorithm is pooling-adjacent-violators (PAV), which optimizes the pooling of adjacent points that violate the monotonicity constraint (Barlow et al., 1972). This initial algorithm has been generalized and incorporated into convex programming frameworks (see Mair et al. (2009) for an excellent summary of the history of isotonic regression and its extensions). Our work builds off of the generalized PAV (GPAV) algorithm of Yu & Xing (2016).

## 3  METHOD

Given inputs $\{\mathbf{x}_i\}_{i=1}^n$, our goal is to learn a latent binary decision tree $\mathcal{T}$ with maximum depth $D$. This tree sends each input $\mathbf{x}$ through branching nodes to a specific leaf node in the tree. Specifically, all branching nodes $\mathcal{T}_B \subset \mathcal{T}$ split an input $\mathbf{x}$ by forcing it to go to its left child if $s_\theta(\mathbf{x}) < 0$, and right otherwise. There are three key parts of the tree that need to be identified: 1. The *continuous* parameters $\theta_t \in \mathbb{R}^d$ that describe how $s_{\theta_t}$ splits inputs at every node $t$; 2. The *discrete* paths $\mathbf{z}$ made by each input $\mathbf{x}$ through the tree; 3. The *discrete* choice $a_t$ of whether a node $t$ should be active or pruned. We describe how to learn each of these below.

### 3.1  TREE-TRAVERSAL & PRUNING PROGRAMS

Imagine the splitting functions of the tree $s_{\theta_t}$ are fixed. Given this, the following integer linear program (ILP) describes how inputs $\mathbf{x}$ traverse the tree. The solution $z_{it} \in \{0, 1\}$ indicates if $\mathbf{x}_i$ reaches node $t \in \mathcal{T}$ (for notational simplicity let $\mathbf{z}_i \in \{0, 1\}^{|\mathcal{T}|}$ be the vectorized indicator for $\mathbf{x}_i$),

$$\max_{\mathbf{z}_1, \dots, \mathbf{z}_n} \quad \sum_{i=1}^n \mathbf{z}_i^\top \mathbf{q}_i \tag{1}$$

$$\text{s.t. } \forall i \in [n], \; q_{it} = \min\{r_{it}, l_{it}\}$$
$$r_{it} = \min\{s_{\theta_{t'}}(\mathbf{x}_i), \; \forall t' \in A_R(t), \}$$
$$l_{it} = \min\{-s_{\theta_{t'}}(\mathbf{x}_i), \; \forall t' \in A_L(t)\}$$
$$z_{it} \in \{0, 1\}.$$

Here $A_L(t)$ is the set of ancestors of node $t$ whose left child must be followed to get to $t$, and similarly for $A_R(t)$. The quantities $q_{it}$ (where $\mathbf{q}_i \in \mathbb{R}^{|\mathcal{T}|}$ is the tree-vectorized version of $q_{it}$) describe the 'reward' of sending $\mathbf{x}_i$ to node $t$. This is based on how well the splitting functions leading to $t$ are satisfied ($q_{it}$ is positive if all splitting functions are satisfied and negative otherwise).

Notice that the solution is unique so long as $s_{\theta_t}(\mathbf{x}_i) \neq 0$ for all $t \in \mathcal{T}, i \in \{1, \dots, n\}$ (*i.e.*, $s_{\theta_t}(\mathbf{x}_i) = 0$ means there is no preference to split $\mathbf{x}_i$ left or right). Further note that **the integer constraint on $z_{it}$ can be relaxed to an interval constraint $z_{it} \in [0, 1]$ without loss of generality**. This is because if $s_{\theta_t}(\mathbf{x}_i) \neq 0$ then $z_t = 0$ if and only if $q_t < 0$ and $z_t = 1$ when $q_t > 0$ (and $q_t \neq 0$).

While the above program works for any fixed tree, we would like to be able to also learn the structure of the tree itself. Let $\eta_t \in \mathbb{R}$ be our preference for pruning/keeping node $t$ (larger $\eta_t$ indicates node $t$ should be kept). To encourage this pruning preference while ensuring connectivity, we introduce an additional optimization variable $a_t \in \{0, 1\}$, indicating if node $t \in \mathcal{T}$ is active (if 1) or pruned (if 0).

We may now adapt eq. 1 into the following pruning-aware mixed integer program (MIP):

$$\max_{\mathbf{z}_1,\ldots,\mathbf{z}_n,\mathbf{a}} \quad \sum_{i=1}^{n} \mathbf{z}_i^\top \mathbf{q}_i + \boldsymbol{\eta}^\top \mathbf{a} \tag{2}$$

$$\text{s.t. } \forall i \in [n], \ a_t \le a_{p(t)}, \quad \forall t \in \mathcal{T} \setminus \{\text{root}\}$$
$$z_{it} \le a_t$$
$$z_{it} \in [0,1], a_t \in \{0,1\}.$$

Here we have removed the first three constraints in eq. 1 as they are a deterministic computation independent of $\mathbf{z}_1,\ldots,\mathbf{z}_n,\mathbf{a}$. Further $p(t)$ indicates the parent of node $t$. The added constraint $a_t \le a_{p(t)}$ ensures that child nodes $t$ are pruned if parent nodes $p(t)$ are pruned. While the new constraint $z_{it} \le a_t$ ensures that no point $\mathbf{x}_i$ can reach node $t$ if node $t$ is pruned.

## 3.2 Learning Tree Parameters

A natural approach to learn splitting parameters $\theta_t$ would be to do so in the MIP itself, as in the optimal tree literature. However, this would severely restrict allowed splitting functions as even linear splitting functions can only practically run on at most thousands of training inputs (Bertsimas & Dunn, 2017). Instead, we propose to learn $s_{\theta_t}$ via gradient descent. To do so, we must be able to compute the gradients $\frac{\partial \mathbf{z}}{\partial \boldsymbol{\eta}}$ and $\frac{\partial \mathbf{z}}{\partial \mathbf{q}}$. However, the solutions of eq. 2 are discontinuous and piecewise-constant.

To solve this, we relax the integer constraint on $\mathbf{a}$ to the interval $[0,1]$ and add quadratic regularization $1/2\sum_i \|\mathbf{z}_i\|_2^2 + 1/2\|\mathbf{a}\|_2^2$. Rearranging and negating the objective yields

$$\mathrm{T}_{\boldsymbol{\eta}}(\mathbf{q}_1,\ldots,\mathbf{q}_n) = \underset{\mathbf{z}_1,\ldots,\mathbf{z}_n,\mathbf{a}}{\arg\min} \ 1/2\sum_{i=1}^{n} \|\mathbf{z}_i - \mathbf{q}_i\|^2 + 1/2\|\boldsymbol{\eta} - \mathbf{a}\|^2 \tag{3}$$

$$\text{s.t. } \forall i \in [n], \ a_t \le a_{p(t)}, \quad \forall t \in \mathcal{T} \setminus \{\text{root}\}$$
$$z_{it} \le a_t$$
$$z_{it} \in [0,1], a_t \in [0,1].$$

The regularization makes the objective strongly convex, so from convex duality it follows that $\mathrm{T}_{\boldsymbol{\eta}}$ is Lipschitz continuous (Zalinescu, 2002, Corollary 3.5.11). By Rademacher's theorem (Borwein & Lewis, 2010, Theorem 9.1.2), $\mathrm{T}_{\boldsymbol{\eta}}$ is thus differentiable almost everywhere. Generic methods such as OptNet (Amos & Kolter, 2017) could be used to compute the solution and the gradients. However, by using the tree structure of the constraints, we next derive an efficient specialized algorithm. The main insight, shown below, reframes pruned binary tree learning as isotonic optimization.

**Proposition 1.** *Let* $\mathcal{C} = \{\mathbf{a} \in \mathbb{R}^{|\mathcal{T}|} : a_t \le a_{p(t)} \text{ for all } t \in \mathcal{T} \setminus \{root\}\}$. *Consider*

$$\mathbf{a}^\star = \underset{\mathbf{a} \in \mathcal{C} \cap [0,1]^{|\mathcal{T}|}}{\arg\min} \ \sum_{t \in \mathcal{T}} \left( 1/2(a_t - \eta_t)^2 + \sum_{i:a_t \le q_{it}} 1/2(a_t - q_{it})^2 \right). \tag{4}$$

*Define[2]* $[z^\star]_{it} = \mathrm{Proj}_{[0,a_t^\star]}(q_{it})$. *Then,* $\mathrm{T}_{\boldsymbol{\eta}}(\mathbf{q}_1,\ldots,\mathbf{q}_n) = \mathbf{z}_1^\star,\ldots,\mathbf{z}_n^\star, \mathbf{a}^\star$.

*Proof.* The constraints and objective function of eq. 3 are separable, so we may push the minimization *w.r.t.* $\mathbf{z}$ inside the objective, resulting in:

$$\underset{\mathbf{a} \in \mathcal{C} \cap [0,1]^{|\mathcal{T}|}}{\arg\min} \ 1/2\|\boldsymbol{\eta} - \mathbf{a}\|^2 + \sum_{t \in \mathcal{T}} \sum_{i=1}^{n} \min_{0 \le z_{it} \le a_t} 1/2(z_{it} - q_{it})^2. \tag{5}$$

Each of the inner nested minimizations, $\min_{0 \le z_{it} \le a_t} 1/2(z_{it} - q_{it})^2$ is a one-dimensional projection onto box constraints, with solution $z_{it}^\star = \mathrm{Proj}_{[0,a_t]}(q_{it})$. We may use this result to eliminate $\mathbf{z}$ from the objective, noting that

$$1/2(z_{it}^\star - q_{it})^2 = \begin{cases} 1/2\, q_{it}^2, & q_{it} < 0 \\ 0, & 0 \le q_{it} < a_t \\ 1/2(a_t - q_{it})^2, & q_{it} \ge a_t \end{cases} \tag{6}$$

---

[2] Here $\mathrm{Proj}_{\mathcal{S}}(x)$ is the projection of $x$ onto set $\mathcal{S}$. If $\mathcal{S}$ are box constraints, projection amounts to clipping.

---

**Algorithm 1** Latent decision tree induction via isotonic optimization.

1: initial partition $\mathcal{G} \leftarrow \{\{1\}, \{2\}, \cdots\} \subset 2^{\mathcal{T}}$
2: **for all** $G \in \mathcal{G}$ **do**
3:     $d_G \leftarrow \arg\min_a \sum_{t \in G} g_t(a)$                    ▷ Proposition 2

4: **while** exists $t$ such that $a_t > a_{p(t)}$ **do**
5:     $t_{\max} \leftarrow \arg\max_t \{a_t : a_t > a_{p(t)}\}$
6:     merge $G \leftarrow G \cup G'$ where $t \in G$ and $p(t) \in G'$.
7:     update $a_G \leftarrow \arg\min_a \sum_{t \in G} g_t(a)$              ▷ Proposition 2 again

---

the first two conditions are constants *w.r.t.* $a_t$ Thus, the objective functions eq. (3) and eq. (4) differ by a constant. As their constraints are also the same, they have equivalent minimizers $\mathbf{a}^\star$.     □

**Efficiently inducing trees as isotonic optimization.**     From Proposition 1, notice that eq. 4 is an instance of tree-structured isotonic optimization: the objective decomposes over *nodes*, and the inequality constraints correspond to edges in a rooted tree:

$$\arg\min_{\mathbf{a} \in \mathcal{C}} \sum_{t \in \mathcal{T}} g_t(a_t), \quad \text{where} \quad g_t(a_t) = \frac{1}{2}(a_t - \eta_t)^2 + \sum_{i:a_t \leq q_{it}} \frac{1}{2}(a_t - q_{it})^2 + \iota_{[0,1]}(a_t). \quad (7)$$

where $\iota_{[0,1]}(a_t) = \infty$ if $a_t \notin [0,1]$ and 0 otherwise. This problem can be solved by a generalized *pool adjacent violators* (PAV) algorithm: Obtain a tentative solution by ignoring the constraints, then iteratively remove violating edges $a_t > a_{p(t)}$ by *pooling together* the nodes at the end points. At the optimum, the nodes are organized into a partition $\mathcal{G} \subset 2^{\mathcal{T}}$, such that if two nodes $t, t'$ are in the same group $G \in \mathcal{G}$, then $a_t = a'_t := a_G$.

When the inequality constraints are the edges of a rooted tree, as is the case here, the PAV algorithm finds the optimal solution in at most $|\mathcal{T}|$ steps, where each involves updating the $a_G$ value for a newly-pooled group by solving a one-dimensional subproblem of the form (Yu & Xing, 2016)[3]

$$a_G = \arg\min_{a \in \mathbb{R}} \sum_{t \in G} g_t(a), \quad (8)$$

resulting in Algorithm 1. It remains to show how to solve eq. 8. The next result, proved in Appendix A.1, gives an exact and efficient solution, with an algorithm that requires finding the nodes with highest $q_{it}$ (*i.e.*, the nodes where $\mathbf{x}_i$ is most highly encouraged to traverse).

**Proposition 2.** *The solution to the one-dimensional problem in eq. (8) for any $G$ is given by*

$$\arg\min_{a \in \mathbb{R}} \sum_{t \in G} g_t(a) = \mathrm{Proj}_{[0,1]}\left(a(k^\star)\right) \quad \text{where} \quad a(k^\star) := \frac{\sum_{t \in G} \eta_t + \sum_{(i,t) \in S(k^\star)} q_{it}}{|G| + k^\star}, \quad (9)$$

$S(k) = \{j^{(1)}, \ldots, j^{(k)}\}$ *is the set of indices* $j = (i,t) \in \{1, \ldots, n\} \times G$ *into the $k$ highest values of* $\mathbf{q}$*, i.e.,* $q_{j^{(1)}} \geq q_{j^{(2)}} \geq \ldots \geq q_{j^{(m)}}$*, and $k^\star$ is the smallest $k$ satisfying $a(k) > q_{j^{(k+1)}}$.*

Figure 2 shows the speed of our specialized algorithm compared to a leading generic optimizer (details in Appendix A.2).

**Backward pass and efficient implementation details.**     Algorithm 1 is a sequence of differentiable operations that can be implemented *as is* in automatic differentiation frameworks. However, because of the prominent loops and indexing operations, we opt for a low-level implementation as a `C++` extension. Since the $\mathbf{q}$ values are constant *w.r.t.* $\mathbf{a}$, we only need to sort them once as preprocessing. For the backward pass, rather than relying on automatic differentiation, we make two remarks about the form of $\mathbf{a}$. Firstly, its elements are organized in groups, *i.e.*, $a_t = a'_t = a_G$ for $\{t, t'\} \subset G$. Secondly, the value $a_G$ inside each group depends only on the optimal support set $S_G^\star := S(k^\star)$ as

---

[3]Compared to Yu & Xing (2016), our tree inequalities are in the opposite direction. This is equivalent to a sign flip of parameter $\mathbf{a}$, *i.e.*, to selecting the *maximum* violator rather than the minimum one at each iteration.

---

**Algorithm 2** Learning with latent decision tree representations.

1: initialize neural network parameters $\phi, \theta$
2: initialize pruning scores $\boldsymbol{\eta} \sim \mathcal{U}(-1, 1)$
3: **repeat**
4:     sample batch $\{\mathbf{x}_i\}_{i \in B}$
5:     induce traversals $\{\mathbf{z}_i\}_{i \in B}, \mathbf{a} = \mathrm{T}_{\boldsymbol{\eta}}\big(\{q_{\boldsymbol{\theta}}(\mathbf{x}_i)\}\big)$          ▷ algorithm 1; differentiable
6:     update parameters using $\nabla_{\{\boldsymbol{\theta}, \boldsymbol{\eta}, \phi\}} \ell(f_\phi(\mathbf{x}_i, \mathbf{z}_i)) + \lambda \|\boldsymbol{\eta}\|_\Omega$          ▷ autograd

---

defined for each subproblem by Proposition 2. Therefore, in the forward pass, we must store only the node-to-group mappings and the sets $S_G^\star$. Then, if $G$ is the group of node $t$,

$$\frac{\partial a_t^\star}{\partial \eta_{t'}} = \begin{cases} \frac{1}{|G|+k^\star}, & 0 < a_t^\star < 1 \text{ and } t' \in G, \\ 0, & \text{otherwise.} \end{cases} \qquad \frac{\partial a_t^\star}{\partial q_{it'}} = \begin{cases} \frac{1}{|G|+k^\star}, & 0 < a_t^\star < 1, \text{ and } (i, t') \in S_G^\star, \\ 0, & \text{otherwise.} \end{cases}$$

As $\mathrm{T}_{\boldsymbol{\eta}}$ is differentiable almost everywhere, these expressions yield the unique Jacobian at all but a measure-zero set of points, where they yield one of the Clarke generalized Jacobians (Clarke, 1990). We then rely on automatic differentiation to propagate gradients from $\mathbf{q}$ to the split parameters $\theta$; since $\mathbf{q}$ is defined elementwise via $\min$ functions, the gradient propagates through the minimizing path, by Danskin's theorem (Proposition B.25, Bertsekas, 1999; Danskin, 1966).

### 3.3 THE OVERALL OBJECTIVE

We are now able to describe the overall optimization procedure that simultaneously learns tree parameters: (a) input traversals $\mathbf{z}_1, \ldots, \mathbf{z}_n$; (b) tree pruning $\mathbf{a}$; and (c) split parameters $\theta_t$. Given this tree, we will additionally learn a function $f_\phi(\mathbf{z}, \mathbf{x})$ to minimize an arbitrary loss $\ell(\cdot)$ as follows,

$$\min_{\boldsymbol{\theta}, \boldsymbol{\eta}, \phi} \quad \sum_{i=1}^n \ell\big(f_\phi(\mathbf{x}_i, \mathbf{z}_i)\big) + \lambda \|\boldsymbol{\eta}\|_\Omega \tag{10}$$
$$\text{where} \quad \mathbf{z}_1, \ldots, \mathbf{z}_n, \mathbf{a} := \mathrm{T}_{\boldsymbol{\eta}}\big(q_{\boldsymbol{\theta}}(\mathbf{x}_1), \ldots, q_{\boldsymbol{\theta}}(\mathbf{x}_n)\big).$$

In practice, we perform mini-batch updates for efficient training; the procedure is sketched in Algorithm 2. Recall $\mathrm{T}_{\boldsymbol{\eta}}(\cdot)$ is the solution to the tree program (eq. 3), so we may update the parameters $\boldsymbol{\eta}$ and $\boldsymbol{\theta}$ by back-propagation. Here we define $q_{\boldsymbol{\theta}}(\mathbf{x}_i) := \mathbf{q}_i$ to make explicit the dependence of $\mathbf{q}_i$ on $\theta$. The regularization $\|\cdot\|_\Omega$ can be any norm (in practice we find $\Omega = \infty$ to perform the best.) The overall model is represented in Figure 1.

## 4 EXPERIMENTS

In this section we showcase our method on both: (a) *Classification/Regression* for tabular data, where tree-based models have been demonstrated to have superior performance over MLPs (Popov et al., 2019); and (b) *Hierarchical clustering* on unsupervised data. Our experiments demonstrate that our method leads to predictors that are competitive with state-of-the-art tree-based approaches. Further we visualize the trees learned by our method, and how sparsity is easily adjusted by tuning the regularization parameter $\lambda$.

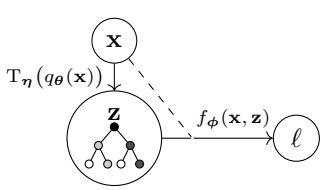

**Figure 1:** Model overview.

**Architecture details.**   We use a linear function or a multi-layer perceptron ($L$ fully-connected layers with Elu activation (Clevert et al., 2015) and dropout) for $f_\phi(\cdot)$ and choose between linear or linear followed by Elu splitting functions $s_\theta(\cdot)$ (we limit the search for simplicity, there are no restrictions except differentiability).

### 4.1 SUPERVISED LEARNING ON TABULAR DATASETS

Our first set of experiments is on supervised learning with heterogeneous tabular datasets, where we consider both regression and binary classification tasks. We minimize the Mean Square Error (MSE)

**Table 1:** Results on tabular datasets. We report average and standard deviations over 4 runs of MSE for regression datasets: *Year*, *Microsoft* and *Yahoo*, and error rate for classification datasets: *Click* and *Higgs*. Best result, and those within a standard deviation from it, for each family of algorithms (single tree or ensemble) are in **bold**. Dashes '-' indicate that the method cannot be run (i.e., a method is designed only for classification). Experiments are run on a machine with 16 CPUs and 63GB of RAM, with a training time limit of 3 days. We denote methods that exceed this memory and training time as OOM and OOT, respectively.

| | method | **YEAR** | **MICROSOFT** | **YAHOO** | **CLICK** | **HIGGS** |
|---|---|---|---|---|---|---|
| Single Tree | CART | $97.40 \pm 0.37$ | $0.6914 \pm 3\text{e-}3$ | $0.6190 \pm 3\text{e-}3$ | $0.3780 \pm 6\text{e-}3$ | $0.3220 \pm 1\text{e-}3$ |
| | DNDT | - | - | - | $0.4866 \pm 1\text{e-}2$ | OOM |
| | OPTREE | - | - | - | $0.4100 \pm 4\text{e-}2$ | OOT |
| | NDF-1 | - | - | - | $\mathbf{0.3344 \pm 5\text{e-}4}$ | $0.2644 \pm 8\text{e-}4$ |
| | ANT | $77.84 \pm 0.55$ | $\mathbf{0.5721 \pm 17\text{e-}4}$ | $\mathbf{0.5893 \pm 22\text{e-}4}$ | | |
| | Ours | $\mathbf{77.11 \pm 0.21}$ | $\mathbf{0.5728 \pm 4\text{e-}4}$ | $\mathbf{0.5911 \pm 5\text{e-}4}$ | $\mathbf{0.3342 \pm 4\text{e-}4}$ | $\mathbf{0.2227 \pm 6\text{e-}4}$ |
| Ens. | NODE | $\mathbf{76.21 \pm 0.12}$ | $0.5570 \pm 2\text{e-}4$ | $0.5692 \pm 2\text{e-}4$ | $\mathbf{0.3312 \pm 2\text{e-}3}$ | $\mathbf{0.2101 \pm 5\text{e-}4}$ |
| | XGBoost | $78.53 \pm 0.09$ | $\mathbf{0.5544 \pm 1\text{e-}4}$ | $\mathbf{0.5420 \pm 4\text{e-}4}$ | $\mathbf{0.3310 \pm 2\text{e-}3}$ | $0.2328 \pm 3\text{e-}4$ |

on regression datasets and the Binary Cross-Entropy (BCE) on classification datasets. We compare our results with tree-based architectures, which either train a single or an ensemble of decision trees. Namely, we compare to the greedy classification and regression trees (CART) (Breiman et al., 1984) and to the optimal decision tree learner with local search (Optree-LS; Dunn, 2018). We also consider three baselines with probabilistic routing: deep neural decision trees (DNDT; Yang et al., 2018), deep neural decision forests (Kontschieder et al., 2015) configured to use an ensemble size of 1 (NDF-1) and adaptive neural networks (ANT; Tanno et al., 2019). As for the ensemble baselines, we compare to NODE (Popov et al., 2019), the state-of-the-art method for training a forest of differentiable oblivious decision trees on tabular data, and to XGBoost (Chen & Guestrin, 2016), a scalable tree boosting method. We carry out the experiments on the following datasets. **Regression:** *Year* (Bertin-Mahieux et al., 2011), Temporal regression task constructed from the Million Song Dataset; *Microsoft* (Qin & Liu, 2013), Regression approach to the MSLR-Web10k Query–URL relevance prediction for learning to rank; *Yahoo* (Chapelle & Chang, 2011), Regression approach to the C14 learning-to-rank challenge. **Binary classification:** *Click*, Link click prediction based on the KDD Cup 2012 dataset, encoded and subsampled following Popov et al. (2019); *Higgs* (Baldi et al., 2014), prediction of Higgs boson–producing events.

For all tasks, we follow the preprocessing and task setup from (Popov et al., 2019). All datasets come with training/test splits. We make use of 20% of the training set as validation set for selecting the best model over the epochs and for tuning the hyperparameters. We tune the hyperparameters for all methods but for Optree, as its long training time makes hyper-parameter tuning unfeasible. Details are provided in the appendices. Finally, we optimize eq. (10) and all neural network methods (DNDT, NDF, ANT and NODE) using the Quasi-Hyperbolic Adam (Ma & Yarats, 2018) stochastic gradient descent method, with default parameters and batch size equal to 512. Table 1 reports the obtained results.[4] Unsurprisingly, ensemble methods outperfom single-tree ones on all datasets, although at the cost of being harder to visualize/interpret. Our method has the advantage of (a) generalizing to any task; (b) outperforming or matching all single-tree methods; (c) approaching the performance of ensemble-based methods. Additional experiments reported in the appendices show that our model is also significantly faster to train, compared to its differentiable tree counterparts NDF-1 and ANT, while matching or beating the performance of these baselines.

**Further comparison with optimal tree baselines.** We run a set of experiments on small binary classification datasets to compare our method with optimal tree methods. Specifically we compare against two versions of Optree: one that solves the MIP exactly (Optree-MIP) (Bertsimas & Dunn, 2017), and another that solves it with local search Optree-LS (Dunn, 2018). We also compare with the state-of-the-art optimal tree method of Lin et al. (2020), called GOSDT. We consider the *Mushrooms* binary classification dataset (Schlimmer, 1987): prediction between edible and poisonous mushrooms, with 8124 instances and 22 features We apply a stratified split on both datasets to obtain 60%-20%-20% training-validation-test sets, convert categorical features to ordinal, and z-score them. For our method, we apply the Quasi-Hyperbolic Adam optimization algorithm, with batch size equal to 512. Further details about the experimental setup are available in the appendices.

---

[4]DNDT, Optree-LS and NDF handle only classification tasks.

**Table 2:** Results on the Mushrooms tabular dataset. We report average training time (s), and average and standard deviations of test error rate over 4 runs for binary classification datasets. We **bold** the best result (and those whose standard deviation overlaps the mean of the best result). Experiments are run on a machine with 16 CPUs and 63GB RAM. We limit the training time to 3 days.

| | MUSHROOMS | |
|---|---|---|
| method | training time (s) | error rate |
| OPTREE-MIP | OOT | - |
| OPTREE-LS | OOT | - |
| GOSDT | 214 | $\mathbf{0.0122 \pm 0.0027}$ |
| Ours | **66** | $0.0723 \pm 0.0544$ |

**Table 3:** Results for hierarchical clustering. We report average and standard deviations of *dendrogram purity* over four runs. Best results and all within a standard deviation from it are in bold.

| METHOD | GLASS | COVTYPE |
|---|---|---|
| Ours | $\mathbf{0.494 \pm 0.025}$ | $\mathbf{0.467 \pm 0.006}$ |
| gHHC | $0.463 \pm 0.002$ | $0.444 \pm 0.005$ |
| HKMeans | $\mathbf{0.508 \pm 0.008}$ | $0.440 \pm 0.001$ |
| BIRCH | $0.429 \pm 0.013$ | $0.440 \pm 0.002$ |

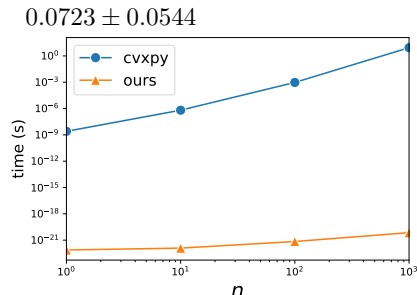

**Figure 2:** Comparison of Algorithm 1's running time (ours) with the running time of cvxpy (Diamond & Boyd, 2016; Agrawal et al., 2018) with the OSQP solver (Stellato et al., 2020) for tree depth $D=3$, varying $n$.

Table 2 reports the performance of all methods across 4 runs. Both OPTREE variants do not finish in under 3 days. On *Mushrooms* our method runs 3 times faster than GOSDT at the expense of higher error. We believe the slow scaling of GOSDT is due to the fact that it binarizes features, working with 118 final binary attributes on *Mushrooms*.

## 4.2 SELF-SUPERVISED HIERARCHICAL CLUSTERING

To show the versatility of our method, we carry out a second set of experiments on hierarchical clustering tasks. Inspired by the recent success of self-supervised learning approaches (Lan et al., 2019; He et al., 2020), we learn a tree for hierarchical clustering in a self-supervised way. Specifically, we regress a subset of input features from the remaining features, minimizing the MSE. This allows us to use eq. (10) to learn a hierarchy (tree). To evaluate the quality of the learned trees, we compute their dendrogram purity (DP; Monath et al., 2019). DP measures the ability of the learned tree to separate points from different classes, and corresponds to the expected purity of the least common ancestors of points of the same class.

We experiment on the following datasets: *Glass* (Dua & Graff, 2017): glass identification for forensics, and *Covtype* (Blackard & Dean, 1999; Dua & Graff, 2017): cartographic variables for forest cover type identification. For *Glass*, we regress features 'Refractive Index' and 'Sodium,' and for *Covtype* the horizontal and vertical 'Distance To Hydrology.' We split the datasets into training/validation/test sets, with sizes 60%/20%/20%. Here we only consider linear $f_\phi$. As before, we optimize Problem 10 using the Quasi-Hyperbolic Adam algorithm, with batch size equal to 512 for *Covtype* and 8 for *Glass*, and tune the hyper-parameters using the validation set.

As baselines, we consider: BIRCH (Zhang et al., 1996) and Hierarchical KMeans (HKMeans), the standard methods for performing clustering on large datasets; and the recently proposed gradient-based Hyperbolic Hierarchical Clustering (gHHC) (Monath et al., 2019) designed to construct trees in hyperbolic space. Table 3 reports the dendrogram purity scores for all methods. Our method matches or outperforms all methods, even though not specifically tailored to hierarchical clustering.

**Tree Pruning**   The hyper-parameter $\lambda$ in eq. 10 controls how aggressively the tree is pruned, hence the amount of tree splits that are actually used to make decisions. This is a fundamental feature of our framework as it allows to smoothly trim the portions of the tree that are not necessary for the downstream task, resulting in lower computing and memory demands at inference. In Figure 3, we

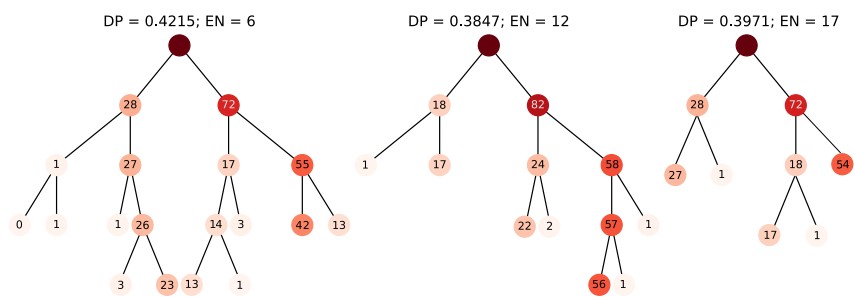

**Figure 3:** *Glass* tree routing distribution, in rounded percent of dataset, for $\lambda$ left-to-right in $\{0, 0.001, 0.1\}$. The larger $\lambda$, the more nodes are pruned. We report dendrogram purity (DP) and total empty nodes (EN).

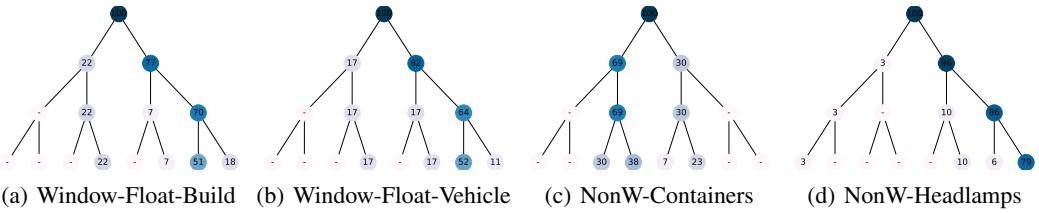

(a) Window-Float-Build   (b) Window-Float-Vehicle   (c) NonW-Containers   (d) NonW-Headlamps

**Figure 4:** Class routing distributions on *Glass*, with distributions normalized over each depth level. Trees were trained with optimal hyper-parameters, (depth $D=6$), but we plot nodes up to $D=4$ for visualization ease.

study the effects of pruning on the tree learned on *Glass* with a depth fixed to $D=3$. We report how inputs are distributed over the learned tree for different values of $\lambda$. We notice that the number of empty nodes, *i.e.*, nodes that are not traversed by any data point, increases together with $\lambda$ up to a certain value in order not to significantly deteriorate results (as measured by dendrogram purity).

**Class Routing** In order to gain insights on the latent structure learned by our method, we study how points are routed through the tree, depending on their class. The *Glass* dataset is particularly interesting to analyze as its classes come with an intrinsic hierarchy, *e.g.*, with superclasses *Window* and *NonWindow*. Figure 4 reports the class routes for four classes. As the trees are constructed without supervision, we do not expect the structure to exactly reflect the class partition and hierarchy. Still, we observe that points from the same class or super-class traverse the tree in a similar way. Indeed, trees for class *Build* 4(a) and class *Vehicle* 4(b), which both belong to *Window* super-class, share similar paths, unlike the classes Containers 4(c) and Headlamps 4(d).

## 5 DISCUSSION

In this work we have presented a new optimization approach to learn trees for a variety of machine learning tasks. Our method works by sparsely relaxing a ILP for tree traversal and pruning, to enable simultaneous optimization of these parameters, alongside splitting parameters and downstream functions via argmin differentiation. Our approach nears or improves upon recent work in both supervised learning and hierarchical clustering. We believe there are many exciting avenues for future work. One particularly interesting direction would be to unify recent advances in tight relaxations of nearest neighbor classifiers with this approach to learn efficient neighbor querying structures such as ball trees. Another idea is to adapt this method to learn instance-specific trees such as parse-trees.

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

## A  APPENDIX

### A.1  PROOF OF PROPOSITION 2

Let $G \subset \mathcal{T}$ be a subset of (pooled) node indices. We seek to solve

$$\operatorname*{arg\,min}_{a \in \mathbb{R}} \sum_{t \in G} g_t(a) \;:=\; \operatorname*{arg\,min}_{a \in [0,1]} \sum_{t \in G} {}^{1}\!/{}_{2}(a - \eta_t)^2 + \sum_{i : a \leq q_{it}} {}^{1}\!/{}_{2}(a - q_{it})^2 \tag{11}$$

Note that the final summation implicitly depends on the unknown $a$. But, regardless of the value of $a$, if $q_{it} \leq q_{i't'}$ and $q_{it}$ is included in the sum, then $q_{i't'}$ must also be included by transitivity. We may therefore characterize the solution $a^\star$ via the number of active inequality constraints $k^\star = \left| \{ (i,t) : a^\star \leq q_{i,t} \} \right|$. We do not know $a^\star$, but it is trivial to find by testing all possible values of $k$. For each $k$, we may find the set $S(k)$ defined in the proposition. For a given $k$, the candidate objective is

$$J_k(a) = \sum_{t \in G} {}^{1}\!/{}_{2}(a - \eta_t)^2 + \sum_{(i,t) \in S(k)} {}^{1}\!/{}_{2}(a - q_{it})^2 \tag{12}$$

and the corresponding $a(k)$ minimizing it can be found by setting the gradient to zero:

$$J_k'(a) = \sum_{t \in G}(a - \eta_t) + \sum_{(i,t) \in S(k)}(a - q_{i,t}) := 0 \iff a(k) = \frac{\sum_{t \in G} \eta_t + \sum_{(i,t) \in S(k)} q_{it}}{|G| + k}. \tag{13}$$

Since $|S(k)| = k$ and each increase in $k$ adds a non-zero term to the objective, we must have $J_1\big(a(1)\big) \leq J_1\big(a(2)\big) \leq J_2\big(a(2)\big) \leq \ldots$, so we must take $k$ to be as small as possible, while also ensuring the condition $|\{(i,t) : a(k) \leq q_{it}\}| = k$, or, equivalently, that $a(k) > q_{j([k+1])}$. The box constraints may be integrated at this point via clipping, yielding $a^\star = \operatorname{Proj}_{[0,1]}\big(a(k^\star)\big)$.

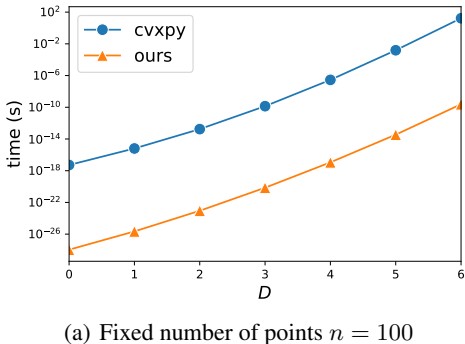 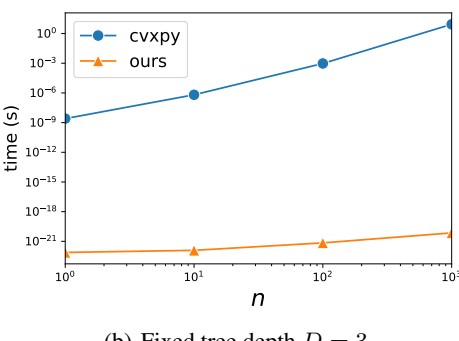

(a) Fixed number of points $n = 100$          (b) Fixed tree depth $D = 3$

**Figure 5:** Comparison of Algorithm 1's running time (ours) with the running time of **Cvxpy** with OSQP solver. $n$ takes values in a range that covers common choices of batch size. Time and $n$ axis are represented in logarithmic scale.

## A.2  BENCHMARKING SOLVER TIMES

We study the running time of Algorithm 1 depending on the number of data points $n$ and the chosen tree depth $D$. We compare its solving time with the time needed by the popular convex optimization framework **Cvxpy** (Diamond & Boyd, 2016; Agrawal et al., 2018) to solve Problem 3 with OSQP solver (Stellato et al., 2020)[5]. As Cvxpy is based on solvers implemented in Objective C and C we implement our approach in C++ for a fair comparison. We report the solving times in Figure 5, where we vary one parameter $n$ or $D$ at a time and fix the other. The algorithm that we specifically designed to solve Problem (3) is indeed several magnitude faster than the considered generic solver.

## A.3  FURTHER EXPERIMENTAL DETAILS

We tune the hyper-parameters of all methods with Optuna Bayesian optimizer (Akiba et al., 2019), fixing the number of trials to 100. For all back-propagation-based methods, we fix the learning rate to 0.001, use a scheduler reducing this parameter of a factor of 10 every 2 epochs where the validation loss has not improved, and fix the maximal number of epochs to 100 and patience equal to 10 for early stopping. For our method, we further initialize the bias $\boldsymbol{b} \in \mathbb{R}^{|\mathcal{T}|}$ for the split function $s_\theta(x)$ (explicitly defining $s_\theta(x) = s_{\theta \setminus \boldsymbol{b}}(x) + \boldsymbol{b}$) to ensure that points are equally distributed over the leaves at initialization. We hence initialize the bias to minus the initial average value of the training points traversing each node: $b_t = -\frac{1}{|\{x_i | q_{it} > 0\}_{i=1}^n|} \sum_{i=1}^n s_{\theta_t \setminus b_t}(x_i) \mathbb{1}[q_{it} > 0]$.

**Experiments on tabular datasets**  The other hyper-parameters for these experiments are chosen as follows:

- **Ours**: we tune $D$ in $\{2, \ldots, 6\}$, $\lambda$ in $[1\text{e-}3, 1\text{e+}3]$ with log-uniform draws, the number of layers of the MLP $L$ in $\{2, \ldots, 5\}$ and its dropout probability uniformly in $[0, 0.5]$, and the choice of activation for the splitting functions as linear or Elu;

- **Optree-LS**: we fix the tree depth $D = 6$;

- **CART**: we tune $D$ in $\{2, \ldots, 10\}$, *feature rate* uniformly in $[0.5, 1]$, *minimal impurity decrease* in $[0, 1]$, $\alpha$ log-uniformly in $[1\text{e-}16, 1\text{e+}2]$ and *splitter* chosen between best or random;

- **Node** and **XGBoost**: all results are reported from Popov et al. (2019), where they used the same experimental set-up;

- **DNDT**: we tune the softmax temperature uniformly between $[0, 1]$ and the number of feature cuts in $\{1, 2\}$;

- **NDF**: we tune $D$ in $\{2, \ldots, 6\}$ and fix the feature rate to 1;

---

[5]We ran experiments with the commercial solver GUROBI (Gurobi Optimization, 2020) but didn't find significant differences with using OSQP.

**Table 4:** Number of parameters for single-tree methods on tabular datasets.

| | method | YEAR | MICROSOFT | YAHOO | CLICK | HIGGS |
|---|---|---|---|---|---|---|
| Single Tree | CART | 164 | 58 | 883 | 12 | 80 |
| | DNDT | - | - | - | 4096 | - |
| | OPTREE-LS | - | - | - | 3060 | - |
| | NDF-1 | - | - | - | 78016 | 47168 |
| | ANT | 179265 | 17217 | 53249 | | |
| | Ours | 15892 | 19158 | 59131 | 239 | 255 |

- **ANT**: for the sake of fairness, we choose as transformer the identity function, as router a linear layer followed by the Relu activation and with soft (sigmoid) traversals, and as solver a MLP with $L$ hidden layers, as defined for our method; we hence tune $L$ in $\{2, \ldots, 5\}$ and its dropout probability uniformly in $[0, 0.5]$, and fix the maximal tree depth $D$ to 6; we finally fix the number of epochs for growing the tree and the number of epochs for fine-tuning it both to 50.

**Experiments on small datasets**  We chose the hyper-parameters as follows:

- **Ours**: we tune $D$ in $\{2, \ldots, 6\}$, $\lambda$ in $[1e\text{-}3, 1e\text{+}3]$ with log-uniform draws, and make use of a linear predictor and of linear splitting functions without activation;

- **Optree-MIP/Optree-LS**: we fix the tree depth to $D = 6$;

- **GOSDT**: we tune the regularization parameter $\lambda$ in $[1e\text{-}3, 1e\text{+}3]$ with log-uniform draws, and set accuracy as the objective function.

**Experiments on hierarchical clustering**  For this set of experiments, we make us of a linear predictor and of linear splitting functions without activation. The other hyper-parameters of our method are chosen as follows: we tune $D$ in $\{2, \ldots, 6\}$, $\lambda$ in $[1e\text{-}3, 1e\text{+}3]$ with log-uniform draws. The results of the baselines are reported from Monath et al. (2019).

## A.4   ADDITIONAL EXPERIMENTS

In Figure 6 we represent the average test Error Rates or Mean Square Error as a function of the training time for each single-tree method on the tabular datasets of Section 4.1. Notice that our method provides the best trade-off between time complexity and accuracy over all datasets. In particular, it achieves Error Rates comparable on *Click* and significantly better on *Higgs* w.r.t. **NDF-1** while being several times faster. Table 4 shows that this speed-up is principally due to a smaller number of model's parameters. Despite having model sizes comparable to ANT's ones on *Microsoft* and *Yahoo*, our method is significantly faster than this baseline as it offers an efficient way for optimizing the tree structure (via the optimization of pruning vector $\boldsymbol{a}$). In comparison, ANT needs to incrementally grow trees in a first phase, to then fine-tune them in a second phase, resulting in a computational overhead.

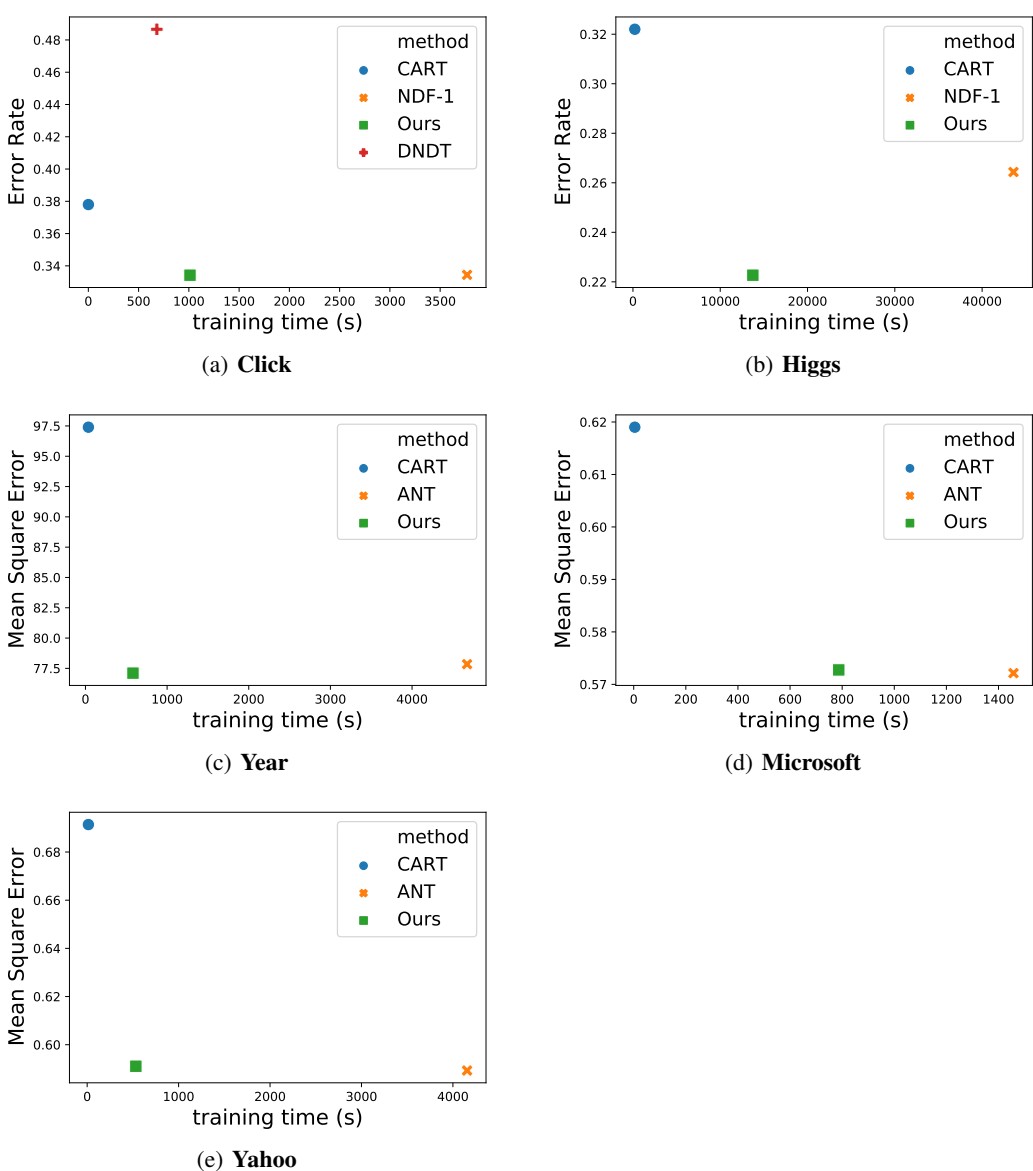

(a) **Click**

(b) **Higgs**

(c) **Year**

(d) **Microsoft**

(e) **Yahoo**

**Figure 6:** Average (a,b) Error Rate (c-e) Mean Square Error vs average training time required by each method.

