# OpenReview forum: "Learning Binary Trees via Sparse Relaxation"
_ICLR.cc/2021/Conference — Reject_

### Official Review · AnonReviewer2 · 2020-10-27
**Quite generic but unclear tree learning algorithm**

**Rating:** 4
**Confidence:** 4

**Review:**

The given paper describes a novel approach of learning binary trees using a differentiable relaxation of MIP. For that matter, authors describe their approach of tree traversal for the given input x and a complete binary tree of depth D. This problem is formulated as MIP which is relaxed to make it differentiable. This MIP is then reformulated to learn a tree structure (i.e. tree pruning). Finally, a decision node parameters are optimized using backpropagation. The algorithm alternates between tree structure learning using relaxed MIP and tree parameters learning using SGD.

Major advantageous:

  . The proposed method is interesting and addresses the tree learning algorithm from different perspective. I liked the idea of MIP relaxation and the proposed algorithm to solve the QP (using isotonic regression).

  . It is nice that the method is quite generic and can be applied to both (semi)supervised/unsupervised (e.g. clustering) problems.

Major concerns:

  . First of all, I found this paper is hard to follow and understand. Therefore, I have a couple of fundamental questions/comments regarding methodology:

    - The overall obj function (eq. 10) is minimized using backpropagation over decision node parameters \theta. However, in order to compute gradients w.r.t. \theta, we need to compute q(x) which is, I believe, non-differentiable (it involves max function).

    - Are you solving the tree program (eq. 3) at each mini-batch update or it is done only once at the beginning of the minimization problem 10? Providing a pseudocode of the overall algorithm would be beneficial.

    - In conventional trees, a tree output is given by its leaf. However, here it seems the final prediction is obtained by f(.) and I don't understand why? What is the use of f(.)? I know that it is a linear function or MLP. But, why it takes z and x as arguments and makes the final prediction. It is not well described in the paper. By the way, figure 1 is not referenced anywhere...

  . Given all relaxations in the final algorithm, my impression is that these trees are look like a regular soft trees but with tree structure learning capability.

  . Experiments lack sufficient evaluations (for regression and classification). For regression, comparison is performed only against CART. Moreover, why depth of the tree for CART is limited to {2,...,6}? I believe that it heavily under-represents CART since it is known that the depth for such trees should be sufficiently large. Most importantly, I strongly suggest authors to include more advanced baselines (e.g. [1,2,3]) for both classification and regression. Moreover, resulting table should contain model sizes in terms of number of parameters, at least (e.g. I'm confident that CART trees will be much smaller).

[1] Tanno, R., Arulkumaran, K., Alexander, D. C., Criminisi, A., and Nori, A. Adaptive neural trees. ICML 2019.

[2] Carreira-Perpinan, M. A. and Tavallali, P. (2018). Alternating optimization of decision trees, with application to learning sparse oblique trees. NeurIPS 2018.

[3] A. Zharmagambetov and M. A. Carreira-Perpinan (2020): Smaller, More Accurate Regression Forests Using Tree Alternating Optimization. ICML 2020

Minor concerns:

  . I believe that the method cannot be used to train a regular axis-algined trees, i.e. to force each node to use only one feature.

  . NDF baseline is always performs better than the proposed approach. I believe ANT[1] will perform even better. Given all of them have similar model sizes (including the proposed method), I don't see any practical benefits of the proposed method over the others.

---

> ### Author Response · Authors · 2020-11-22
> **Clarification on formulation and advantages of proposed method**
>
> 1. However, in order to compute gradients w.r.t. \theta, we need to compute q(x) which is, I believe, non-differentiable (it involves max function).
>
> This is a common misconception, thank you for pointing it out, we have added a clarification at the end of Section 3.2. The max function is differentiable almost everywhere, this is not a problem; in fact, ReLU(z) = max(z, 0) is commonly used as an activation in neural nets; our computation is similar. The gradient of a minimum of a family of functions propagates along the minimizing branch, by Danskin's theorem. (Nonlinear Programming, Bertsekas, 1999, Proposition B.25)
>
> 2. Are you solving the tree program (eq. 3) at each mini-batch update [...]? Providing a pseudocode of the overall algorithm would be beneficial.
>
> We solve the tree program at each mini-batch, as we need to derive the tree representations $z_i$ for all points. Thanks for this suggestion, we have included algorithm pseudocode in the updated version (Algorithm 2).
>
> 3. In conventional trees, a tree output is given by its leaf. However, here it seems the final prediction is obtained by f(.) and I don't understand why?
>
> We agree that this aspect of our work should be highlighted more clearly: instead of making predictions using heuristic scores over the training points assigned to a leaf (e.g., majority class), our approach allows one to learn a prediction function f() that optimizes an overall objective. One way to think of this is that our approach embeds a decision tree as a layer of a deep neural network and optimizes the continuous parameters of the whole model by back-propagation. In our experiments, we considered a simple neural architecture, where z and a final predictor f(x, z) are learned end-to-end. Beyond the benefits of joint optimization, this final predictor is also useful for adapting the tree to certain settings. For example, in regression settings the labels often lie in a large (or even unknown) interval of the reals. The final predictor f() is particularly useful here as it can learn to map the traversals z, which lie in [0, 1], to arbitrary intervals.
>
> 4. Given all relaxations in the final algorithm, my impression is that these trees are look like a regular soft trees but with tree structure learning capability.
>
> The only relaxations we make are to input-traversals and tree-pruning decisions. The relaxation is sparse so most values will be either 0 or 1 and very few will fall within [0,1]. This way our method is different from most differentiable soft tree relaxations such as Irsoy et al., 2012; DNDT; NDF.
>
> 5. Experiments lack sufficient evaluations (for regression and classification) [...] I strongly suggest authors to include more advanced baselines (e.g. [1,2,3]) for both classification and regression.
>
> We have added additional comparisons to optimal tree baselines suggested by R3 in Table 3. We are also currently running [1] and will cite all of the above works.
>
> 6. why depth of the tree for CART is limited to {2,...,6}? I believe that it heavily under-represents CART...
>
> Thank you for catching this. In fact we searched CART depths in {2,...,10}, thank you for drawing attention to this typo! We found empirically that CART trees quickly start to overfit on these datasets as the depth increases. Specifically, for the considered supervised learning datasets, the tree depth selected by hyperparameter tuning was always smaller or equal to 7.
>
> 7. resulting table should contain model sizes in terms of number of parameters[...]
>
> We have updated the paper to include the number of parameters for single-tree methods on supervised learning datasets, in Table 2.. Apart from CART our model is consistently smaller than both optimal tree and differentiable tree methods.
>
> 8. I believe that the method cannot be used to train a regular axis-algined trees[...]
>
> Our method does not learn axis-aligned trees but this is an interesting direction for future work. We believe we could use SparseMax (Martins & Astudillo, 2016) to learn approximately axis-aligned splits, or REINFORCE (Williams, 1988) to learn exact axis-aligned splits.
>
> 9. NDF baseline is always performs better than the proposed approach. I believe ANT[1] will perform even better. Given all of them have similar model sizes (including the proposed method), I don't see any practical benefits of the proposed method over the others.
>
> As pointed out by R3 and R4, our method can be applied to a wide variety of tasks, even to unsupervised problems, which is not the case for NDF and ANT. Moreover, as detailed by R4, it offers an efficient way for optimizing the tree structure (via the optimization of pruning vector a). In comparison, ANT needs to incrementally grow trees. We are currently running ANT on all tabular datasets and we have observed that this approach is significantly slower than ours. Finally, aside from CART, our approach uses fewer parameters than all other large-scale supervised tree learning methods as shown in Table 2.

---

> > ### Comment · AnonReviewer2 · 2020-11-23
> > **Thank you for the clarification**
> >
> > More stuff become clear now. Thanks. Some further questions/comments:
> >
> >  4. The only relaxations we make are to input-traversals and tree-pruning decisions. The relaxation is sparse so most values will be either 0 or 1 and very few will fall within [0,1]. This way our method is different from most differentiable soft tree relaxations such as Irsoy et al., 2012; DNDT; NDF.
> >
> > Afaik, in soft trees, we can traverse each input to all nodes with some probability which is equivalent to having relaxed z in your case, isn't it? Thus I can see a strong similarity between these two methods. However, soft trees are trained end-to-end using gradient based techniques whereas here some sort of alternating optimization involved.
> >
> >  5. We have added additional comparisons to optimal tree baselines suggested by R3 in Table 3. We are also currently running [1] and will cite all of the above works.
> >
> > I believe that the experimental evaluation is still a weak point here: regression is compared against CART only, only 2 relatively simple classification datasets were used. I guess the choice of datasets mainly driven from (Popov et al., 2019). I suggest to include more relevant datasets (e.g. from [2,3], mnist could be a very nice benchmarks since it is a well studied) and compare against baselines therein.

---

### Official Review · AnonReviewer4 · 2020-10-28
**Good paper**

**Rating:** 7
**Confidence:** 3

**Review:**

Summary: The paper provides an interesting way to learn binary trees that is faster than generic solvers of MIP. The trick is to use gradient-based methods for a new sparse relaxation of MPI. Authors compare their method to generic solvers and find a substantial improvement on runtime. This is certainly important for scalability.

The paper overall is of good quality. The story of the work is well-written which makes the contributions easier to digest.

In terms of theoretical contributions, the work is "weak" in the sense that the main result follows from a simple relaxation. However, authors show strong empirical evidence that their method is faster and competitive to existing results. Thus, in terms of scalability, the results are relevant.

If my understanding is correct, the appealing aspect of their method is the runtime. In the experiments section, authors show a huge gap in runtime between cvxpy and their implementation. I wonder if cvxpy is mostly implemented in Python, given that the authors implemented their method in C++, I would expect a better performance just from that. Although, the huge gap in runtime suggests that it wouldn't matter.

The topic is also of good significance given that decision trees are still widely used in practice.

---

> ### Author Response · Authors · 2020-11-22
> **Scalability and clarification on comparison with CVXPY**
>
> 1.  I wonder if cvxpy is mostly implemented in Python, given that the authors implemented their method in C++, I would expect a better performance just from that. Although, the huge gap in runtime suggests that it wouldn't matter.
>
> Thanks for bringing this up. CVXPY is based on three open source solvers that are implemented largely in C and Objective C, which is why we implemented our method in a similarly fast language for comparison. We have added this to the appendix to make this clearer.
>
> Overall we’d like to thank the reviewer for emphasizing one of the major advantages of our method. Specifically, not only is the proposed relaxation and the derived algorithm for inducing tree traversals several magnitudes faster than state-of-the-art techniques, but it also makes the formulation differentiable, allowing our method to learn trees using extremely efficient automatic differentiation libraries. This further allows us to embed a decision tree as a module of a deep learning architecture and optimize it with arbitrary objectives, even in unsupervised settings.

---

### Official Review · AnonReviewer1 · 2020-10-31
**The experimental results are promising but writing lacks important information**

**Rating:** 3
**Confidence:** 3

**Review:**

The problem definitions are not somewhat unclear. It seems that (1)-(7) is to find a pruning of the given tree, which are not explicitly mentioned.

It is not nontrivial that the problem (1) leads to a tree. It would be necessary to have some proposition that ensures the desired properties (e.g., solution represents a tree) of the solution or cites such previously known ones.

The obtained solutions in (3)-(7) are not discrete and how to round up continuous solutions to discrete ones are not shown.

As a summary, the current writing lacks important information such as the correctness of the solutions, which significantly reduces the contribution of the paper, even if the experimental results look better than previous work.

---

> ### Author Response · Authors · 2020-11-22
> **Clarification about formulation**
>
> 1. The problem definitions are not somewhat unclear. It seems that (1)-(7) is to find a pruning of the given tree, which are not explicitly mentioned.
>
> We would like to clarify our formulation. The solution to Eq. (1) characterizes tree routing. As mentioned right above it, Eq. (2) extends (1) to jointly model tree routing and pruning. Eq. (3) is a relaxed and smooth (thus differentiable) version of (2). Eqs. (4-7) describe a customized way to solve the optimization problem (3).
>
> 2. It is not nontrivial that the problem (1) leads to a tree...
>
> Problem (1) is essentially a simplification of the MIP in Bertsimas & Dunn, 2017 in Eq. (24). The proof that it yields a tree is as follows. For any x_i, fix a tree depth and all splitting functions $s_{\theta_t}$. Given this tree there exists only one leaf node $l^*$ for which itself and all its parents have $s_{\theta_t}$ values greater than $0$. This is because for any branch at node $t$, $x_i$ splits right if $s_{\theta_t}(x_i) > 0$ and left if $-s_{\theta_t}(x_i) > 0$. Thus, there is one path that $x_i$ can follow where $s_{\theta_t}(x_i) > 0$ for all considered nodes $t$, until hitting the leaf node $l^*$. Finally for any $t$, $q_{it}$ records the minimum value along the path to reach $t$ for $x_i$. Thus, $q_{it}$ is negative for all nodes along all paths except the one leading to $l^*$. Thus $z$ will be positive only for the nodes leading to $l^*$ and $0$ otherwise.
>
> 3. The obtained solutions in (3)-(7) are not discrete and how to round up continuous solutions to discrete ones are not shown...
>
> We do not perform any rounding. If discrete solutions are necessary one could simply round to the nearest integer.

---

### Official Review · AnonReviewer3 · 2020-10-31
**Recommending Weak Accept.**

**Rating:** 6
**Confidence:** 4

**Review:**

## Summary

This paper presents a new approach to learn decision trees via sparse relaxation. The approach starts from a mixed-integer program that can simultaneously induce and prune optimal decision trees from data. The proposed technique aims to solve a continuous relaxation of this problem by combining (1) a tree induction routine, which uses isotonic optimization with (2) an efficient implementation that avoids the need for automatic differentiation. The paper includes experiments on publically available datasets, showing how the decision trees produced via sparse relaxation to decision trees produced using other tree induction approaches.

## Pros

- Interesting new technique for a canonical problem.
- Proposed method produces trees that appear to perform well on classification, regression, and clustering problems.
- Paper showcases a deft approach to modern algorithm design

## Cons

- No theory or empirical results to characterize the optimality of trees produced via sparse relaxation.

## Rating

I have awarded the paper a 6, though I am open to increasing my score if the authors can address some of the questions I include below. Overall, my chief concern about this work is that it does not include any evidence pertaining to the optimality for trees produced via sparse relaxation. In effect, the main argument for the proposed sparse relaxation technique is that it produces trees that perform well on five datasets. I believe that my concerns should be easy to address – either by reporting the optimality gap of the trees in the experimental results, or by including comparisons with methods that are guaranteed to find optimal trees.


##  Questions / Comments

1. Have you evaluated the optimality of solutions of your method with respect to the exact tree-fitting optimization problem – i.e., problem (10) where the tree is a solution to the MIP in (2). All decision trees should be feasible with respect to this problem. In turn, it should be possible to report their "optimality gap." Reporting the optimality gap would provide some evidence to evaluate the integrity of the decisions you made in algorithm decision. It could also showcase the value of sparse relaxation as a feasibility heuristic for the MIP-based approach.

2. In the experiments, what is the relative optimality gap of the solutions found by OPTREE? Is it finding certifiably optimal solutions (i.e., solutions with a relative optimality gap of 0%)? If not, it would be useful to include comparisons on a tabular dataset that is small enough for OPTREE to find a certifiably optimal solution. Two datasets to consider here are (1) UCI Mushrooms dataset (which is linearly separable and should be easy for all methods), and (2) the COMPAS dataset (also small, but not linearly separable).

FWIW, the statement that "Bertsimas & Dunn (2017) phrased the objective of CART as a MIP that could be solved exactly" is misleading as it suggests that Bertsimas & Dunn (2017) are able to recover globally optimal solutions to the MIP. This is not the case. In many datasets, the proposed method can only find feasible solutions that perform well but have large optimality gaps.

3. I am wondering why the authors did not consider the following methods to train optimal decision trees in their experiments:

- [Optimal Sparse Decision Trees](https://papers.nips.cc/paper/8947-optimal-sparse-decision-trees)
- [Generalized and Scalable Optimal Sparse Decision Trees](https://proceedings.icml.cc/static/paper_files/icml/2020/3364-Paper.pdf)

This seems like a weird oversight given that: (i) the paper includes references to both of these works; (ii) both works include reliable implementation that could easily be used to train decision trees in the experimental section (see e.g., https://github.com/Jimmy-Lin/GeneralizedOptimalSparseDecisionTrees).

The paper currently references both of these works in a way that suggests that they are MIP-based methods. This is misleading. These methods are able to fit certifiably optimal decision trees through specialized algorithms that do not require solving a MIP. I recognize that there has to be a limit to the number of experiments that one can perform. That being said, the trees produced by these methods are an important baseline to include in the experiments since they appear to outperform those produced by MIP-based approaches.

---

> ### Author Response · Authors · 2020-11-22
> **Relation and comparison with methods to train optimal decision trees**
>
> 1. Have you evaluated the optimality of solutions of your method with respect to the exact tree-fitting optimization problem – i.e., problem (10) where the tree is a solution to the MIP in (2)...
>
> We agree this is an interesting question, but determining this optimality gap is highly non-trivial as it would require: 1. Determining the gap between an MIP similar to eq (2) which also includes the variables $\theta$, $\eta$, $\phi$ and its sparse relaxation (which would be a variant of eq (3)); and 2. Arguing about how close the local minima found by stochastic gradient methods are to the global minimum when solving eq (10). While 1 seems possible (we believe there are arguments akin to total unimodularity that could be used for the MIP->QP setting), 2 is at the forefront of non-convex optimization theory and has no clear answers for our setting. We think this is an interesting direction for future work.
>
> 2. FWIW, the statement that "Bertsimas & Dunn (2017) phrased the objective of CART as a MIP that could be solved exactly" is misleading...
>
> Thank you for the clarification, we will rephrase.
>
> 3. I am wondering why the authors did not consider the following methods to train optimal decision trees in their experiments...
>
> While this line of work is indeed an important influence, we stress that there is a significant difference in our motivation, as we compromise global tree optimality in order to allow learning within a neural network. We did try this comparison, and on large datasets it is not as easy as claimed, for computational reasons. On the datasets reported in the initial submission, GSOSDT (Lin et al., 2020) ran out of memory. At your suggestion, we have run OPTREE (Bertsimas & Dunn, 2017) and GOSDT on the smaller Mushrooms and COMPAS datasets, and reported the results in the updated version of the paper. Our method achieves error rates comparable to those of GOSDT and it is several times faster on these small-scale datasets. We haven’t been able to obtain OPTREE’s performance yet, either the MIP version (Bertsimas & Dunn, 2017) or the local search version (Dunn, 2018), after 3 days of training per run. This is currently running and we will report the results as soon as we get them. Overall, we want to stress that our method is aimed at a different setting than these methods: learning flexible trees for arbitrary objectives.

---

### Author Response · Authors · 2020-11-22
**We thank all reviewers**

We would like to thank all reviewers for their time and thoughtful comments. We appreciated the suggestions for improving the paper and updated it accordingly.
In the official comments, we address each reviewer's concerns point by point.

---

### Comment · Area_Chair1 · 2020-11-24
**Boosting vs MIP**

One question comes to mind for a Boosting person: the work of Kearns and Mansour (started w/ STOC 1996: On the Boosting Ability of Top-Down Decision Tree Learning Algorithms) and follow ups (see the citing papers) show that greediness has a purpose in the boosting framework -- it gives provable and fast convergence under lightweight assumptions about splits. However,  the convergence rate depends on the splitting criterion and CART's criterion displays the "poorest" of all. I understand the technical interest in picking Gini index in the paper's context. However, Kearns and Mansour give an optimal splitting criterion -- not the binary entropy of C4.5 but another one with possibly better technical appeal in the context of the paper. Could it be used ?

I also suspect it would be possible, at least until section 3.1, to analyze the boosting abilities of the proposed method following Kearns and Mansour's  framework.

---

### Author Response · Authors · 2020-11-25
**Comments on submitted revised paper**

We submitted a revised version of the paper with the following updates:
1. We revised the related work as suggested by reviewers;
2. We clarified our formulation and its derivation;
3. We added a pseudocode for the overall optimization procedure in Algorithm 2, following reviewer 2’s suggestion;
4. We reported the results of Adaptive Neural Trees (ANT) on regression datasets in Table 1, to address reviewer 2’s concerns about the limited number of baselines in this setting. We will update the paper with ANT’s performance on classification datasets as well as soon as the tuning and training are complete (these have yet to finish after 7 days);
5. We improved our method’s results, by initializing the split functions so that points are equally spread over the tree at initialization as described in the appendices;
6. We added a comparison with optimal tree baselines in Table 2, as suggested by reviewer 3;
7. We reported training times vs error on the tabular datasets for each baseline in Figure 6 of the appendices, to show that our model is consistently faster to train than existing differentiable tree learning techniques (e.g. NDF, ANT) while achieving state-of-the-art performance;
8. We reported model sizes in Table 4 in the appendices, following reviewer 2’s remarks.

About the comparison to optimal tree baselines of **Table 2**, the current version of the paper reports results only for Mushrooms dataset. As the previous version revealed, on **COMPAS** our method matches the performance of GOSDT with 13 times faster runtime. However **we chose to remove this dataset from our current version of the paper because of the deep ethical concerns posed by the recidivism prediction task**. This is especially important since the reviewers have already filled in ethics reports for our paper based on the initial version that didn’t include this dataset. We are currently working on comparing on other small datasets instead.

---

### Decision · Program_Chairs · 2021-01-07
**Final Decision**

**Decision:**

Reject

**Comment:**

The main problem as flagged by reviewers is the lack of formal evidence that the approach is a right one to carry out. Decision tree induction has early been the subject of formal studies in ML, whether in statistics (Friedman et al.) or ML (Kearns et al.). It is a bit sad that a new approach that relies on a much different standpoint on the problem and modelling of tree classification (Section 3, R2), with experimental results recognized by reviewers (R3, R4) is not accompanied by formal analyses on par with SOTA for related approaches (R3, R1). I would strongly suggest the authors fit in a few more Lemmata, either to follow up on specific problems (R1). The paper would tremendously benefit from extensive connections with the existing theory, be it from the generalization and overfitting standpoint (R2, remark #6) or the choice of the appropriate best contender using the boosting literature. Decision was taken not to accept the paper but I would very strongly encourage the authors to revise the draft.